# The Effectiveness of Virtual Fencing of Bull Calves in a Holistic Grazing System

**DOI:** 10.3390/ani13050917

**Published:** 2023-03-02

**Authors:** Søren Krabbe Staahltoft, Magnus Fjord Aaser, Jakob Nødgaard Strange Jensen, Ismat Zadran, Emil Birkmose Sørensen, Anders Esbjerg Nielsen, Aage Kristian Olsen Alstrup, Dan Bruhn, Anne Cathrine Linder, Christian Sonne, John Frikke, Cino Pertoldi

**Affiliations:** 1Department of Chemistry and Bioscience, Aalborg University, Fredrik Bajers Vej 7H, 9220 Aalborg, Denmark; 2Department of Nuclear Medicine and PET, Aarhus University Hospital, Palle Juul-Jensens Boulevard 165, 8200 Aarhus, Denmark; 3Department of Clinical Medicine, Aarhus University, Palle Juul-Jensens Boulevard 165, 8200 Aarhus, Denmark; 4Skagen Bird Observatory, Fyrvej 36, 9990 Skagen, Denmark; 5National Institute of Aquatic Resources, Technical University of Denmark, Kemitorvet 201, 2800 Kongens Lyngby, Denmark; 6Department of Ecoscience, Aarhus University, Frederiksborgvej 399, 4000 Roskilde, Denmark; 7Wadden Sea National Park, Havnebyvej 30, 6792 Rømø, Denmark; 8Aalborg Zoo, Mølleparkvej 63, 9000 Aalborg, Denmark

**Keywords:** animals, virtual fencing, grazing management, Nofence©, cattle, holistic management

## Abstract

**Simple Summary:**

Virtual fencing is a new way of enclosing livestock without the use of physical barriers. The system relies on GPS technology and works by deterring escapes using auditory warnings and electric impulses. This study examines the effectiveness of one such system in a rotational strip grazing regime with 17 Angus bull calves. This study also aimed to determine which bull calves were more likely to receive electric impulses. The system proved effective at containing the bull calves within the designated enclosure. The bull calves learned to associate the auditory warning with the electric impulse, and as such, received increasingly fewer impulses as time went on. However, in accordance with previous studies, there were clear differences in the number of warnings and impulses received between individuals. The results regarding which bull calves received the most impulses did not show any clear trends. In conclusion, the virtual fencing system proved to be effective and the animals learnt how to interact with the system while receiving very few impulses.

**Abstract:**

Large grazers are essential for nature conservation. In order to prevent grazers from moving to unintended areas, it may be necessary to keep them inside enclosures. Physical fences present a number of problems, such as fragmenting the landscape. Virtual fencing, however, is a possible replacement for physical fencing, making it possible to enclose grazers without physical boundaries. Virtual fencing systems utilise collars with GPS technology to track animals and deliver auditory warnings and electric impulses to keep animals within predefined boundaries. This study examines how effective the virtual fencing system Nofence© is at enclosing calves in a holistically managed setting. Holistic management is a rotational grazing technique where an enclosure is grazed in small strips at a time. It is investigated whether the calves become habituated to the virtual fence and whether there is a correlation between the number of warnings received by every two calves in order to explore potential herd behaviour. Finally, this study examines which calves interact the most with the virtual fence by investigating the relationship between physical activity and number of interactions. Seventeen calves were fitted with a GPS collar from the company Nofence© and placed in a holistically managed enclosure. Data were gathered from 4 July to 30 September 2022. The study found that virtual fence was able to contain calves inside the designated enclosure, and over time the calves received notably fewer electrical impulses compared to auditory warnings. The results of Pearson’s correlation between auditory warnings received by two random calves were inconclusive, but the use of a sliding window analysis should be further explored. Lastly, the most physically active animals were the ones who received the most auditory warnings, but they did not receive more impulses. No significant correlation was found between the number of electric impulses received and the physical activity of the animals.

## 1. Introduction

Conservation of natural areas using grazers is required to reduce the loss of biodiversity [1,2]. This often necessitates the use of physical fencing to keep grazers inside the desired area, as well as to prevent unwanted human encounters and traffic accidents [3]. However, physical fencing presents a number of problems [3,4]. Physical fences can thus have negative impacts on wildlife by creating artificial barriers in the landscape and limiting the free movement of other animals [3,5]. These barriers can also lead to collisions, such as when low-flying bird species collide with fences with lethal consequences [3,6,7,8,9]. An example of an affected species, which are comparable in size and ecology to native European red deer and fallow deer, are the North American white-tailed deer and mule deer, as described by Harrington et al., Burkholder et al. and Bishop et al. [7,10,11]. These papers found that deer were prone to getting caught and dying when attempting to cross fences, with fence crossing success rates at around 75% and mortality rates upwards of 0.40/km/year for pasture fences [7,11]. Moreover, physical fencing can be expensive to install and maintain [5]. Even when fencing is used for nature conservation purposes, it can create new problems when trying to solve others, as physical fences fragment the landscape by preventing animals from moving around freely [3,5,12]. Additionally, physical fencing poses a logistical and practical problem in terms of relocation [6]. In some areas that are difficult to access or have challenging terrain, installing physical fences may not be feasible [13].

Virtual fencing is a possible alternative to physical fencing, without many of the aforementioned problems of physical fencing [13,14,15,16,17,18]. Virtual fencing is a GPS-based system that utilises auditory warnings and electrical impulses to keep livestock within a designated area. As the animal approaches the virtual boundary, warning signals and electrical impulses are administered by a collar fitted around the neck of each animal. This type of system is yet to be legalised in most countries in Europe, including Denmark, due to earlier concerns regarding animal welfare [6,19,20]. However, several studies suggest that welfare is not compromised when utilising virtual fencing systems and livestock are effectively kept within enclosures [13,14,18,20,21,22,23,24,25,26].

In a previous study on virtual fencing by Aaser et al. [14], it was hypothesised that a small enclosure size and regular moving of the virtual border could improve the effectiveness of virtual fencing systems. The reasoning behind the hypothesis was the cattle’s constant interaction with the boundary, through which they would be reminded of the connection between warnings and impulses [14]. According to the responsible veterinarian, the preliminary results of an experiment with holistically managed cattle on the island of Fanø raised some concern, as the animals received a high number of warnings and impulses compared to animals kept in a larger stationary enclosure. Holistic management is a rotational grazing technique where an enclosure is grazed in small strips at a time. The strip is shifted a few metres at regular intervals, usually daily. To the best of our knowledge, little to no research has been conducted to examine the effectiveness of virtual fencing in a holistically managed setting.

Cattle are, in general, gregarious and social creatures living in herds [27,28,29]. They establish hierarchies, where some of the individuals lead the herd, while others are being lead [30]. As suggested by by Aaser et al. [14], one might hypothesise that putting a collar on only some of the animals may be sufficient to keep the entire herd within a virtual enclosure, as has been shown to be the case for sheep [14,31]. This could potentially help reduce the cost even further.

The aim of this study was to investigate whether the Nofence© virtual fencing system is able to contain bull calves in a holistic management setting, whether the calves display herd behaviour and whether it is possible to predict which animals will interact most with the virtual boundary. To support this aim, the following four hypotheses were tested:

(1) A virtual fence is capable of containing bull calves in a specific area during holistic management for more than 99% of the time. (2) Bull calves will receive fewer electrical impulses compared to warnings over time. (3) There is a positive correlation between the number of warnings received by any two bull calves in any given period. (4) Bull calves with the most physical activity receive the most warnings and impulses.

## 2. Materials and Method

### 2.1. Animals and Location

This study took place on the Danish island of Fanø. Seventeen Angus bull calves aged 9 to 12 months were placed in a holistically managed enclosure of 8.2 hectares (Figure 1 and Appendix A). Some of these calves had previously been indirectly exposed to virtual fencing as their mothers were kept in another virtually fenced enclosure, though the bull calves themselves did not wear collars. The pasture was composed of flat cultural grassland. Due to the close proximity of a somewhat busy road, an electric fence was placed around the perimeter of the pasture. An artificial shelter and a water trough were provided in the northwestern side of the enclosure. Furthermore, the neighbouring farmer to the east had a herd of cattle adjacent to the pasture.

### 2.2. Virtual Fencing System

Each animal was fitted with a cattle collar developed by Nofence©. At the time of the experiment, each collar cost NOK 3050, while the system, regardless of the number of collars, cost a yearly digital service fee of NOK 669 (prices provided by Nofence© on 5 December 2022). The collars were the same as described by Aaser et al. [14]. The collars were capable of registering five different types of messages with different information, which were sent to the official Nofence© server (Table 1).

The position of the animal was logged every 15 min and the activity every 30 min. Activity was measured by a step counter developed for another species (sheep) and given as a unitless number. Whenever an animal received a warning or an electric impulse, or the fence status of the collar changed, this was logged at the time of the event along with the position of the animal. The different fence statuses are outlined in Table 2. With every message, the unique serial number of the collar was included. This number was used to reference the animal to which the collar was fitted. Each animal wore the same collar throughout the experiment.

### 2.3. Experimental Protocol

The experiment began on 4 July 2022 and included seventeen Angus bull calves. Each calf was fitted with a collar and was let onto the western side of the field. During the initial days of the experiment, the calves were restricted to an inner enclosure of about 1 ha by an electric fence (Appendix B). This inner enclosure explains why the calves are located to the west during the beginning of the experiment (Appendix B). The virtual boundary was placed close to the electric fence, allowing the animals to receive auditory warnings when approaching the electric fence. The collars also administered electric impulses if the animals continued to move towards the electric fence after receiving a warning. On 9 July, one side of the internal fence was removed, meaning the calves would have to respect the virtual boundary to stay within the enclosure. On 10 July, the other side of the internal fence was removed, and on 11 July, all internal fences were removed. On 12 July, the enclosure size was doubled to 2 ha based on an assessment of the veterinarian associated with the project. At regular intervals, usually around every 24 h, the virtual enclosure was shifted 10–20 m. The fence was shifted eastwards in a circular motion, anchored at the water trough and resting shelters, until the physical boundary of the field was reached after 15 shifts and the procedure started over from the western boundary again (Appendix A). This method was used in order to maintain a balanced grazing intensity. Data collection began on 4 July 2022 and ended on 30 September 2022, totalling a period of 89 days.

### 2.4. Data and Statistical Analysis

All data were downloaded from the official Nofence© servers. Five different message types, sent by the collars, were given as point data, and the virtual enclosure history was given as a polygon for each enclosure version. The start time and end time of each version of the enclosure was recorded. *Poll* messages were sent every 15 min and recorded the position of the animal, the time and the battery voltage in centivolts. *Seq* messages were sent every 30 min and described the activity of the animal and the solar charge accumulated by the collar in the last 30 min. *Warning* and *Zap* messages were sent whenever a warning or an electric impulse had been delivered to the animal. These messages provided the time and position of the animal. *Warning* messages also provided the length of the warning in milliseconds. *Status* messages were sent whenever the fence status of the collar changed, such as when an animal had escaped the enclosure (see Table 2). A total of 241,163 data points were collected. A total of 151,576 were *Poll* messages, 71,990 were *Seq* messages, 13,906 were *Warning* messages, 2153 were *Status* messages and 1538 were *Zap* messages. The data were subdivided by message type and individual. QGIS [32] was used to plot and sort the data, while all statistical analyses were carried out in R version 4.0.3 [33]. Only non-parametric tests were used, as the data did not follow a normal distribution. The collar fell off once for three of the calves, all in the month of August. One was refitted on the same day, while the other two were refitted after two and six days, respectively.

#### 2.4.1. Effectiveness of Virtual Fencing

To evaluate the effectiveness of the virtual fencing system, the number of *Poll* messages with the fence status *Normal* was compared to the number of *Poll* messages with fence status *MaybeOutOfFence* and fence status *Escaped*:$$\frac{Normal}{Normal+MaybeOutOfFence+Escaped}$$

Messages with the status *MaybeOutOfFence* were included to err on the side of caution with regard to the amount of time animals spent outside the enclosure. To visualise the movement of the animals, heatmaps were created for each version of the enclosure. Heatmaps were created for enclosures that were active for more than 12 h. This was done to exclude “intermediary” enclosures, where only one side of the enclosure had been shifted. The number of times an animal broke out of the enclosure was recorded, as well as the length of each breakout. The length of each breakout was determined as the duration of time between the *Fence Status* of the collar changing from *Normal* to *Escaped* and back to *Normal* again, rounded to the nearest half minute.

#### 2.4.2. Habituation to Virtual Fencing

The animals’ habituation to the virtual fence was investigated by recording the number of auditory warnings and electric impulses received by each animal. The cumulative warnings and impulses were calculated and assessed, as well as the length of warnings over time. Furthermore, the number of electric impulses divided by the number of warnings was calculated over time, and the correlation between warnings and electric impulses was calculated using Pearson’s product–moment correlation ($r_{p}$).

#### 2.4.3. Herd Behaviour

The presence of herd behaviour was investigated by calculating the Pearson’s correlation ($r_{p}$) between the number of warnings received by a given calf and any other calf in an 8 h period (00–08 h, 08–16 h and 16–00 h). This was calculated for every pairwise comparison of the 17 animals, resulting in 136 correlations. A sliding window analysis was performed by repeating this calculation for shorter intervals. The correlation was calculated in an 18 period window, denoting the amount of consecutive 8 h periods included in each sliding window position.

#### 2.4.4. Sliding Window Correlation

To account for possible temporal variations in correlations between the animals, a sliding window correlation was used. A sliding window correlation is a way to assess a time series of data in smaller overlapping segments at a time, thus allowing for analysis of a long time series while still capturing short-range temporal variations in the correlation. In practice, the time series is tested one “window” at a time with a certain window size. The window is then shifted one position and the test is run again. As an example, if a window size of four is selected, the test is first run on data from day 1–4, then day 2–5, day 3–6 and so on. The results of these tests are recorded at the first point in the window, denoted the “sliding window position”. The size of the window is increased until the median *p*-value of all correlations is below 0.05, and this is the included result. This way, more than half of the correlations were significant.

#### 2.4.5. Predicting the Individuals with the Most Interactions with the Fence

To determine whether the animals with the most physical activity received the most warnings, Spearman’s rank correlation ($r_{s}$) was calculated between the summed activity of an animal for the entire period and the summed number of warnings and impulses received by the same animal. The Spearman’s correlation ($r_{s}$) was also calculated for the summed number of warnings against the summed number of impulses the animals had received. Furthermore, the Spearman’s correlation ($r_{s}$) was calculated between the summed number of warnings an individual had received and the same individuals mean correlation to all the other animals, with regard to the number of warnings received in 8-h periods.

## 3. Results

### 3.1. Effectiveness of Virtual Fencing

The virtual fencing system effectively kept the animals within the periodically moving enclosure during the 89-day period of data collection. In total, the animals spent 99.74% of the period inside the enclosure. The created heatmaps also indicated that the animals respected the virtual boundaries and stayed within the enclosure (Figure 2 and Appendix B).

Over the course of the 89-day period, 19 breakout events were recorded. If multiple animals escaped at the same time, this was counted as one event per escaped animal. The median escape time was 2 min and the most number of breakouts by an animal was three, with 12 of the 17 calves breaking out of the enclosure at least once (Figure 3).

### 3.2. Habituation to Virtual Fencing

There was a noticeable variation in the number of warnings received by the animals. The lowest total number of warnings received by a calf was 237 (an average of 2.66 warnings per day), while the highest number received was 1874 (an average of 21.06 warnings per day). The frequency of warnings received by the animals seemed to decrease from late August (Figure 4).

As with the number of warnings, the number of impulses received also varied between individuals. The lowest total number of impulses received by a single calf was 36 (an average of 0.40 impulses per day), while the highest number received was 181 (an average of 2.03 impulses per day). It is worth noting that for all calves, the frequency of impulses was highest at the start of the period and stagnated towards the end of the data collection period (Figure 5).

During the the start of the experiment, the warning duration ranged from approximately 0.01 s to 65 s, with a wide spread and many outliers. Within the first couple of weeks, the spread reduced, as well as the median warning duration, which settled around 8–10 s (Figure 6).

The median number of impulses an animal received per warning in the beginning was approximately 0.5, but immediately trended downwards before reaching a median of 0 in late August (Figure 7).

The correlation between the number of daily warnings and impulses an animal received throughout the period showed considerable variation, with the weakest correlation being $r_{p}=0.43$ (ID: 90754) and the strongest being $r_{p}=0.85$ (ID: 89495).

### 3.3. Herd Behaviour

The results of the correlation and sliding window correlation tests run in regards to analysing herd behaviour were inconclusive and have therefore been moved to Appendix C.

### 3.4. Predicting the Number of Warnings for Each Animal

A moderate positive correlation between the summed activity and summed number of warnings an animal received throughout the data collection period was observed ($r_{s}=0.56$, p<0.05). No correlation between summed activity and summed impulses was found. Equally, no significant correlation was found between the summed number of warnings a bull calf had received and the mean correlation between the number of warnings received by the same calf and all other calves.

## 4. Discussion

### 4.1. Effectiveness of Virtual Fencing

Throughout the 89-day period, the virtual fence successfully kept the 17 calves inside the virtual enclosure zone 99.74% of the time, even with daily changes in the virtual boundary. This supports Hypothesis (1), which states that the system will contain the animals more than 99% of the time. This is comparable to another study, where 12 dairy cows were kept within a 0.83 hectare enclosure for 99% of the time in a six day period [25]. The positional data of the calves clearly showed that they respected the virtual boundary (Appendix B). The calves spent most of their time near the sheltering facilities and the water trough located in the northwest of the enclosure. Moreover, the calves grazed close to the virtual boundary to the east, where fresh grass was available (Appendix B). The fact that the calves clearly respected the virtual boundary despite the higher feed availability outside the boundary corresponds with results from other studies where virtual fencing kept cattle away from feed attractants [22,26]. A total of 19 breakouts were recorded between 12 of the animals; however, the fact that the median time before an escaped animal returned to the enclosure was just 2 min could indicate that the breakouts were not deliberate attempts to escape (Figure 3). The remaining five animals did not escape at all. According to the farmer, the calves would sometimes push each other outside the virtual boundary, which could explain some of the breakouts. The ease with which the animals were able to re-enter the enclosure after a breakout, due to the lack of a physical barrier, meant that the farmer did not have to shoo the escaped animals back into the enclosure. The gregarious nature of the bull calves could also explain the short time between an escape and the animal re-entering the enclosure, as they would seek to return to the herd [34]. The five animals that did not escape the enclosure throughout the entire experiment were also the animals that had received the fewest number number of impulses (Figure 3 and Figure 5). This could be an indication that these animals were less prone to be pushed around by their herdmates, or were simply less inquisitive regarding the enclosure border.

### 4.2. Habituation to Virtual Fencing

The frequency of warnings and impulses received by the animals reduced over time, with the number being notably lower in September than in July (Figure 4 and Figure 5). This indicates that the bull calves needed fewer warnings to learn where the virtual boundary was placed, despite the fact that it was shifted at regular intervals. However, in line with several other studies, the amount of warnings and impulses received varied considerably between individuals [14,22,25,26,35]. Equally, the Pearson’s correlation between the number of daily warnings and impulses received varied between individuals (Section 3.2). This variation could be an expression of different personalities among the bull calves or the hierarchical structure of the herd. Some calves continued to receive a relatively high number of warnings, albeit at a lower frequency, while others barely received any warnings in the latter half of the data collection period. This could indicate that those receiving many warnings are “testers” that determine where the new virtual boundary is placed for the rest of the herd. However, this hypothesis is not immediately supported by the observation that, throughout the data collection period, the correlation between the number of warnings received by any two calves was positive for all pair-wise comparisons (Figure A19). If negative correlations had been observed, this would clearly indicate that when these “testers” received warnings, the rest of the herd would actively avoid warnings. However, even if some animals tried to avoid warnings because they observed others interact with the virtual boundary, this would likely be masked by the gregarious nature of cattle. As such, even if they actively tried to avoid warnings by observing others, they would still receive more warnings whenever the herd moved closer to the virtual boundary, making positive correlations very likely, especially due to the narrowness of the virtual enclosure. Despite the inter-individual differences, it is clear that the bull calves became habituated to the virtual fencing system. As such, the amount of impulses compared to warnings decreased over time, which supports Hypothesis (2) (Figure 7). This hypothesis states that the animals would receive fewer impulses per warning over time. This indicates that the animals were increasingly able to react to the auditory warnings to avoid receiving an electric impulse. The median amount of impulses per warning even reached zero in late August. This is positive, as it clearly shows that the bull calves learnt to associate the warnings with impulses, and reacted favourably when receiving warnings. Another study indicated that increased stress does not occur among cows managed with virtual fencing [21]. The decreasing trend in impulses per warning is in line with other studies, which found that the number of warnings and total impulses decreased significantly over time [16,36]. The decrease in the median length of warnings supports the notion that the calves became habituated to the system and reacted to the warnings, turning away from the virtual boundary before receiving an impulse (Figure 6). The warning duration stabilised after a few weeks, which could mean the animals became so familiar with the system that they learnt how long they could safely wait before turning away from the virtual boundary without receiving an impulse.

### 4.3. Herd Behaviour

As a new way of analysing temporal data gathered from animals, such as from these calves, a sliding window correlation was run on the number of warnings any two calves had received in subsequent 8 h periods. A sliding window correlation makes it possible to capture any variations or trends in time, rather than relying on a single result independent of time. In this study, it was decided that due to the large window size of 8 h, the results would be too inaccurate and any discussion of these too speculative. Therefore, the results are only included in Appendix C. The authors of this paper suggest that future studies should continue to explore the use of sliding window correlations and other sliding window analyses to better capture temporal changes and trends.

### 4.4. Predicting the Number of Warnings for Each Animal

A moderate correlation between summed activity and summed number of warnings received was found (Section 3.4). This supports Hypothesis (4), that animals with higher activity receive more warnings. Animals that are more active tend to encounter the virtual fence more frequently than those that are more sedentary. This was expected, as an animal has to be moving to receive a warning. The fact that the correlation was only moderate may be explained by other confounders, such as risk tolerance, foraging and herd behaviour. However, no correlation was found between summed activity and summed number of impulses, indicating that overall activity is not a contributing factor in receiving more impulses (Section 3.4). This does not support our Hypothesis (4), that the most active calves receive the most impulses. Interestingly, the correlation between summed warning and activity and the lack of correlation between impulses and activity indicate that animals that move around more do not receive more impulses than more sedentary animals, despite receiving more warnings. It is possible that the more active animals are better at reacting to the warnings, as they receive them more frequently. It could also simply be that all the animals had learnt to avoid impulses, such that only the number of warnings was affected by the activity of the animal. Further research is required to ascertain which statement is most likely. Lastly, a moderate correlation between summed number of warnings and summed number of impulses was found, which was to be expected, as an animal cannot receive an impulse without first receiving a warning. The correlation is not strong because the animals can receive warnings without impulses. The correlation does support the notion that the animals are learning to react to the warnings when the lack of correlation between activity and impulses is taken into account. A correlation between warnings and impulses would indicate that more active animals receive more impulses due to receiving more warnings, but, as discussed, that is not the case.

## 5. Conclusions

In conclusion, the Nofence© virtual fencing system was capable of keeping the bull calves in a holistically managed enclosure for more than 99% of the time, and the calves that escaped returned of their own accord after a short time. The bull calves became habituated to the virtual fencing system, receiving fewer warnings and impulses over time. Additionally, they learnt to respond favourably to auditory warnings, receiving fewer impulses compared to warnings over time, reaching a median of zero impulses per warning for the last month of the data collection period. The median warning length decreased at the beginning of the period but then stabilised, indicating that the calves became so used to the system that they did not react to warnings immediately, but had figured out when to turn away from the boundary to avoid impulses. A sliding window correlation was used to test for short range changes or temporal trends in the median correlation between the number of warnings received by any two calves, but the results were inconclusive. Some short-range fluctuations were shown using the analysis, showcasing a need for future studies to take temporal variations into account when analysing these types of data. The calves with the highest physical activity received the most warnings, but no other significant results were found when trying to predict which animals would receive the most warnings.

## Figures and Tables

**Figure 1 animals-13-00917-f001:**
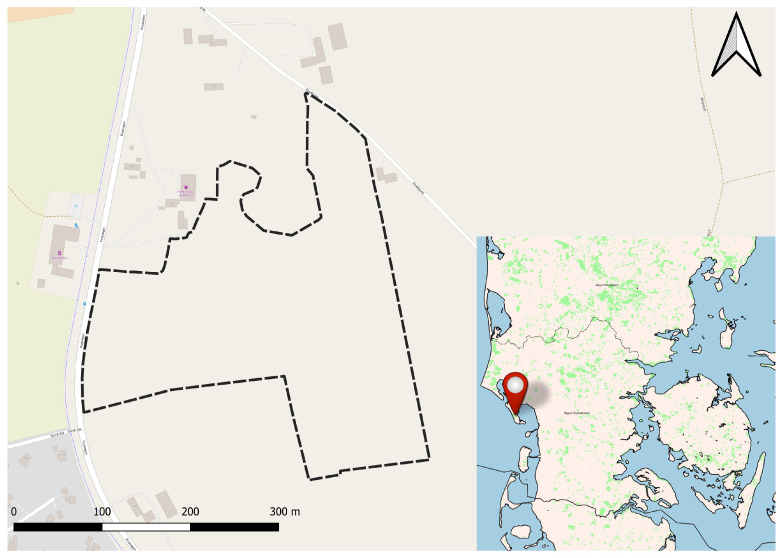
The island of Fanø located in the Wadden Sea of Denmark. The dashed black line represents the physical fence surrounding the 8.2 ha pasture. The coordinates of the upper left corner of the map frame are 55°25′23.131″ E 8°23′53.531″ N. This dataset includes Intellectual Property from European National Mapping and Cadastral Authorities and is licensed on behalf of these by EuroGeographics. Original dataset is available for free at https://www.mapsforeurope.org. Terms of the licence available at https://www.mapsforeurope.org/licenc All attribution statements can be found here.

**Figure 2 animals-13-00917-f002:**
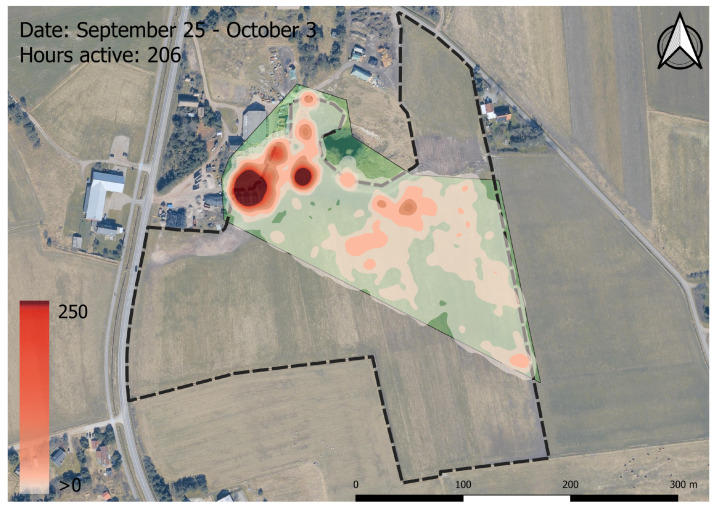
Heatmap visualising all *Poll* messages sent during the period from 25 September to 3 October. The green area shows the virtual enclosure area to which the bull calves were limited. The white to red colour scale indicates the number of *Poll* messages within a 10 m radius on a discrete scale, ranging from 1 point to above 250 points. The coordinates of the upper left corner of the map frame are 55°25′23.131″ E 8°23′53.531″ N. Contains data from the Agency for Data Supply and Infrastructure, Forårsbilleder Ortofoto-GeoDanmark, December 2022.

**Figure 3 animals-13-00917-f003:**
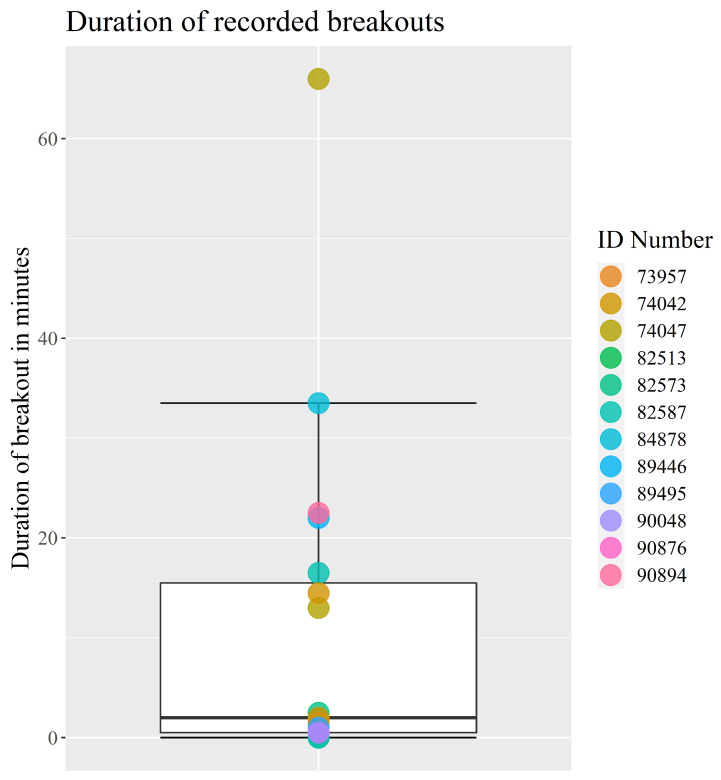
Boxplot of the duration of time spent outside the virtual enclosure for each breakout event during the 89-day period. The points represent each breakout and the colours indicate the escaped individual’s ID number. The boxplot shows min, Q1, median, Q3 and max. Outliers are calculated as 1.5 times IQR. The points are mainly clustered between 0 and 2 min with a few breakouts ranging from 13 to 33.5 min and a single breakout being an outlier at 66 min.

**Figure 4 animals-13-00917-f004:**
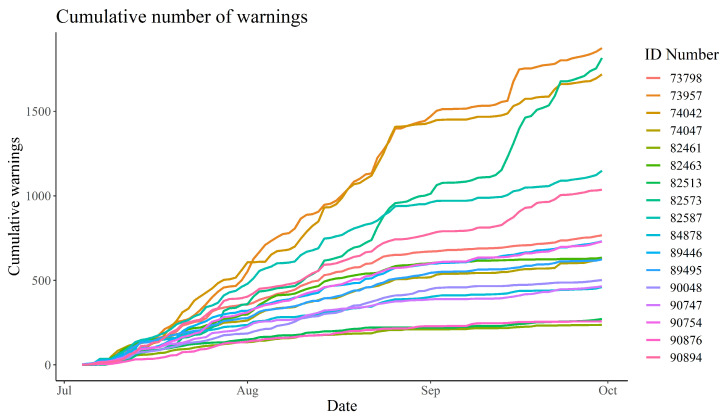
Cumulative number of warnings received by each bull calf between 4 July 2022 and 30 September 2022. The ID Number refers to each of the 17 calves.

**Figure 5 animals-13-00917-f005:**
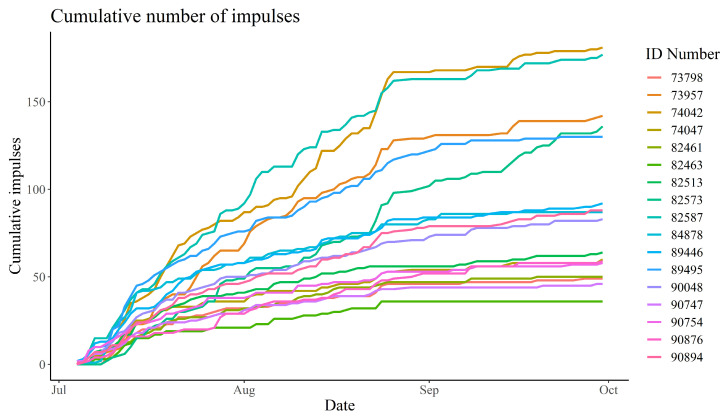
Cumulative number of impulses received by each bull calf between 4 July 2022 and 30 September 2022. The ID Number refers to each of the 17 calves.

**Figure 6 animals-13-00917-f006:**
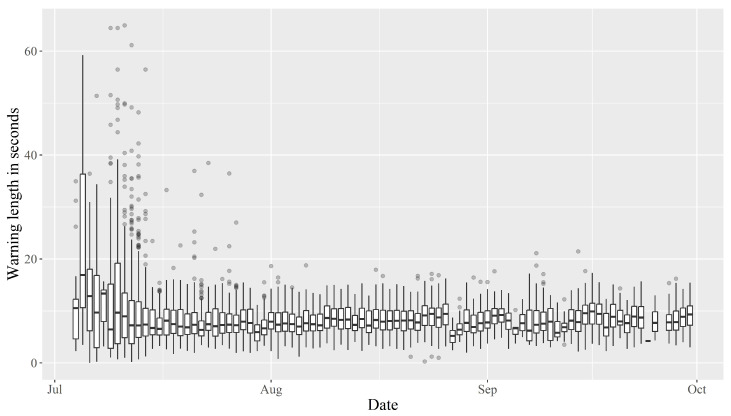
Warning duration in seconds over time for all animals represented as a series of boxplots showing min, Q1, median, Q3 and max. Outliers calculated as 1.5 times IQR.

**Figure 7 animals-13-00917-f007:**
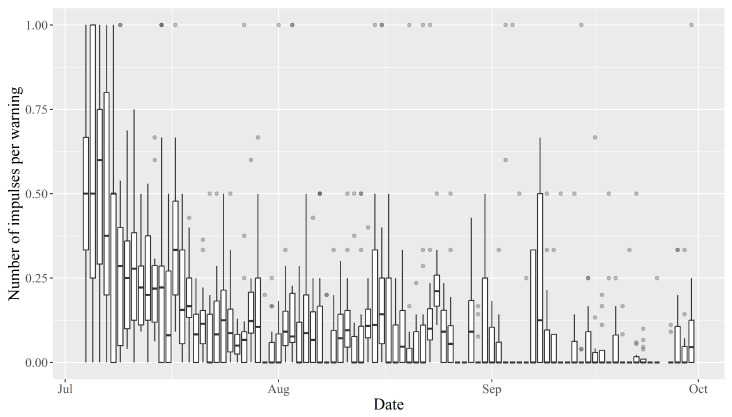
Number of impulses per warning over time. Ratio shown as boxplots showing min, Q1, median, Q2 and max. Outliers calculated as 1.5 times IQR.

**Table 1 animals-13-00917-t001:** Overview of the different message types and description of frequency, data type and number of observations.

Type	Frequency	Data Type	No. of Obs.
*Poll*	Every 15 min	Positional	151,576
*Seq*	Every 30 min	Activity	71,990
*Warning*	After receiving warning	Positional	13,906
*Status*	Upon status change (see Table 2)	Status	2153
*Zap*	Upon receiving impulse	Positional	1538

**Table 2 animals-13-00917-t002:** Overview and description of the different fence statuses.

Fence Status	Description
*Normal*	The virtual enclosure is active
*Not Started*	The virtual enclosure is not active
*Maybe Out Of Fence*	The collar position is outside the enclosure, but the GPS accuracy is not adequate to a trigger warning
*Escaped*	The animal has escaped the enclosure

## Data Availability

The data presented in this study are available on request from the corresponding author.

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
