# Peer review of "The Effectiveness of Virtual Fencing of Bull Calves in a Holistic Grazing System"

_animals, 2023, doi:10.3390/ani13050917_

Round 1

Reviewer 1 Report

The manuscript entitled “The effectiveness of virtual fencing of bull calves in a holistic grazing system” submitted by Staahltoft et al for publication in Animals in the Special Issue Ruminant Welfare Assessment, presented an interesting study. The authors studied the effectiveness of virtual fencing on 17 bull calves for 89 days. Virtual fencings seem to have the potential to become very popular, increase the flexibility to adapt to changing pasture conditions and provide more options for grazing management, especially in high nature value areas. This technology should be further developed, in order to assess animal welfare and lower costs. Not all the land contexts are suitable for such technology, i.e. if large predators are abundant. What about conflicts with large carnivores in the case of virtual fencing? An island has peculiar characteristics and differs from the mainland, please specify if the experiment is suitable for other habitats and how.

The topic is interesting and the data are appropriately presented, with an adequate number of figures, thus prompting to consider the manuscript worth of publication.

Some concerns, It would be useful to have from the authors some more details and compare the chosen virtual fencing system to others in commerce as Vence, Halter, eShepherd etc. see, comment and add in the References Goliński et al 2022. Please specify why Nofence was chosen. I suggest asking the authors to improve the discussion and final conclusions, after some further consideration.

Detailed point-by-point comments, are listed in the following.

Line 26-28: Provide examples and citations on negative impacts on wildlife and particularly on which species, present on the island?

Line 32: landscape fragmentation provide examples

Line 42: the authors need to better specify how animal welfare was assessed, the extent of acoustic deterrence (dB) and voltage of the electric impulse.

Line 74: all the claves where male? If yes why no female was chosen. I think that a table with all the 17th individuals specifying age, morphology and other available information for each animals would help to better understand the paper. Give some more information on the cattle breed.

Figure 1: in order to see the island zoom the map and use a empty square box on the small map and two line to connect the zoomed area.

Line 83: provide approximate price of collar and infrastructure.

Line 90 give more details on warning and electric impulse

Line 189: only 19 breakout recorded seems a very low number for 17 individuals in 89 days need to be discussed and all the possible explanations in the discussion section should be provided.

Line 192: provide in the discussion all the possible explanations on why five individuals never break out from enclosure. Are these individuals younger or of a smaller size (indicate which one are these five in the individual tables I recommended to add)

Author Response

We have not considered large predators as there are almost no large predators in Denmark. The aim of this study was to assess the efficacy of this virtual fencing system, and the focus was on small enclosure size and moving the virtual fence often. To gain a full picture of virtual fencings effectiveness other factors like the presence of large carnivores should be considered, however that was not part of this study.

The experiment did take place on an island, but the enclosure was not located on the coast, and the island is very comparable to the rest of Denmark.

Line 26-28 and Line 32: We have added an example and references on the negative impact of fencing and fragmentation on ungulates that are comparable to the local wildlife.

Line 42: The way animal welfare is assessed differs in the provided references as animal welfare is not a quantifiable metric. The specific method is described in each article. The extend of acoustic warning and voltage is included in the description of the collars referenced in the material section 2.2.

Line 74: The calfs were male because that was the animals that were available from the farmer to participate in the study. We are uncertain whether this information is relevant. We do not believe that a table is necessary as all the 17 animals were of comparable age (9-12 months) and had almost identical living up until the experiment. The breed is described as Angus.

Figure 1: We may be uncertain what you mean, in our understanding the proposed solution does not seem necessary.

Line 83: Cost of collars and digital services from NOFENCE has been provided.

Line 90: The specifics of how the animals receive warnings and impulses is described in the article by Aaser et al. referenced in material section 2.2.

Line 189 and 192: In section 4.1 of the discussion we have mentioned that we believe that 19 breakouts were a large number not a small number. We have changed the wording as we are uncertain whether 19 breakouts is a large or small amount and we have not been able to find statistics on the breakout rate of cattle. Furthermore we have a hard time finding explanations as to why there was not more breakouts, as the cattle had many reasons to leave the enclosure (More food, other animals nearby) and not many reasons other than the virtual fence to stay inside the enclosure. If you have any thoughts on this we would be happy to consider them. We have not considered the why 5 individuals did not escape, as the individuals are all of very similar size and age and there is not really any factors that differentiate the animals. We did notice however that the 5 animals that did no escape were also the animals that had received the fewest impulses and have added a section of discussion on it.

Reviewer 2 Report

My compliments for a well-written article. 

I have two questions/suggestions: 

Abstract: I am not familiar with the term 'holistic management'. I would suggest to use the term 'strip grazing', which is a more familiar term I believe, or explain 'holistic management' in the abstract. It is now explained for the first time in Line 263. 

Introduction: The third hypothesis ("There is a positive correlation between the number of warnings received by two random bull calves in any given period") is not clear to me. Reading further in the article, I understand that it is a measure for herd behaviour. Could you explain this or reformulate this hypothesis? What are you testing, to what aim? 

Author Response

Thank you for your compliment.

We have included a short description of holistic management in the abstract.

We have also made a slight adjustment to hypothesis 3.

Reviewer 3 Report

This is a very good scientific publication that provides new insights into the reactions of grazing animals to virtual fences. Appendices A and B are intended to illustrate the conditions described above. It is doubtful whether this will increase the information content for the reader. These appendices could be dispensed with.

Author Response

Thank you for the compliment.

We agree that the appendices is not necessary information, but we do believe that could be interesting and have therefore placed them in the appendices.